# Diagnostic Biomarkers in Liver Injury by Drugs, Herbs, and Alcohol: Tricky Dilemma after EMA Correctly and Officially Retracted Letter of Support

**DOI:** 10.3390/ijms21010212

**Published:** 2019-12-27

**Authors:** Rolf Teschke, Axel Eickhoff, Amy C. Brown, Manuela G. Neuman, Johannes Schulze

**Affiliations:** 1Department of Internal Medicine II, Division of Gastroenterology and Hepatology, Klinikum Hanau, Academic Teaching Hospital of the Medical Faculty, Goethe University Frankfurt/Main, D-63450 Hanau, Germany; eickhoff.axel@gmx.de; 2Department of Complementary and Integrative Medicine, University of Hawai’i at Manoa, Honolulu, HI 96813, USA; amybrown@hawaii.edu; 3Department of Pharmacology and Toxicology, University of Toronto, Toronto, ON M2 R1 W6, Canada; m_neuman@rogers.com; 4Institute of Occupational, Social and Environmental Medicine, Goethe-University Frankfurt/Main, D-60590 Frankfurt/Main, Germany; j.schulze@em.uni-frankfurt.de

**Keywords:** biomarkers, liver injury, drug induced liver injury, herb induced liver injury, alcoholic liver injury, alcoholic liver disease

## Abstract

Liver injuries caused by the use of exogenous compounds such as drugs, herbs, and alcohol are commonly well diagnosed using laboratory tests, toxin analyses, or eventually reactive intermediates generated during metabolic degradation of the respective chemical in the liver and subject to covalent binding by target proteins. Conditions are somewhat different for idiosyncratic drug induced liver injury (DILI), for which metabolic intermediates as diagnostic aids are rarely available. Although the diagnosis of idiosyncratic DILI can well be established using the validated, liver specific, structured, and quantitative RUCAM (Roussel Uclaf Causality Assessment Method), there is an ongoing search for new diagnostic biomarkers that could assist in and also confirm RUCAM-based DILI diagnoses. With respect to idiosyncratic DILI and following previous regulatory letters of recommendations, selected biomarkers reached the clinical focus, including microRNA-122, microRNA-192, cytokeratin analogues, glutamate dehydrogenase, total HMGB-1 (High Mobility Group Box), and hyperacetylated HMGB-1 proteins. However, the new parameters total HMGB-1, and even more so the acetylated HMGB-1, came under critical scientific fire after misconduct at one of the collaborating partner centers, leading the EMA to recommend no longer the exploratory hyperacetylated HMGB1 isoform biomarkers in clinical studies. The overall promising nature of the recommended biomarkers was considered by EMA as highly dependent on the outstanding results of the now incriminated biomarker hyperacetylated HMGB-1. The EMA therefore correctly decided to officially retract its Letter of Support affecting all biomarkers listed above. New biomarkers are now under heavy scrutiny that will require re-evaluations prior to newly adapted recommendations. With Integrin beta 3 (ITGB3), however, a new diagnostic biomarker may emerge, possibly being drug specific but tested in only 16 patients; due to substantial remaining uncertainties, final recommendations would be premature. In conclusion, most of the currently recommended new biomarkers have lost regulatory support due to scientific misconduct, requiring now innovative approaches and re-evaluation before they can be assimilated into clinical practice.

## 1. Introduction

Shortcomings of correct diagnoses are published for Internal Medicine and viewed as a crucial major problem [1] and are also relevant specifically for Hepatology [2,3] regarding drug induced liver injury (DILI) and herb induced liver injury (HILI) [4]. These challenges require efforts to close gaps between expectations and true medical care quality as suggested by experts in the field [5,6,7,8]. Expectations in clinical medicine are high, asking for excellent diagnoses as the basis for appropriate therapies. In general, diagnosis in internal medicine commonly starts as a symptom based approach. Symptoms have to be analyzed and correctly interpreted, considering specifically the target organ(s) and potential disease(s). Theoretically, biomarkers could assist to establish a correct diagnosis. 

For example, in office settings where a patient presents symptoms of pollakiuria, the physician will focus on the urinary tract using urine tests and may identify a urinary infection or diabetes mellitus as tentative causes before a specific therapy can be initiated. Conditions may be more complex, for instance, in a patient with jaundice, where the physician must exclude a hemolytic disease by using laboratory tests, and the focus will shift to hepatobiliary disorders requiring ultrasound and liver tests (LTs). If a liver disease is likely, a broad range of differential diagnoses is to be considered by the physician [8,9]. Complex liver diseases such as autoimmune hepatitis (AIH) require a specific AIH diagnostic algorithm [10], whereas DILI and HILI need specific diagnostic algorithms including the RUCAM (Roussel Uclaf Causality Assessment Method) [8,9]. Diagnostic biomarkers play an essential role for various liver diseases including HBV and HCV infections [8,9] and are also under discussion for cases of DILI [11,12,13,14] and HILI [15,16]. Unfortunately, regulatory recommendations to use some new biomarkers in clinical trials were officially retracted due to overt misconduct in external studies. 

The focus of this review is on potential diagnostic biomarkers of DILI and HILI with their advantages and shortcomings and emerging issues caused by EMA’s retraction of support leading to a challenging dilemma. In addition, biomarkers to diagnose liver injury by alcohol are discussed as example of liver injury due to other chemicals with hepatotoxic potency.

## 2. Literature Search and Source 

The PubMed database (1964–19 October 2019) was searched for the following terms: biomarkers; drug induced liver injury (DILI); herb induced liver injury (HILI); and alcoholic liver disease (ALD). Key words were used alone or in combination. Limited to the English language, the first 50 publications from each search were analyzed. The final compilation consisted of original papers, consensus reports, and review articles.

## 3. Definitions

### 3.1. Liver Injury

The classic liver injury caused by drugs and herbs is defined by thresholds of alanine aminotransferase (ALT) and alkaline phosphatase (ALP), with serum activities considered as relevant for ALT ≥ 5× ULN (upper limit of normal) and/or ALP ≥2× ULN [8,9,17]. Liver injury is either idiosyncratic, due to the interaction between the exogenous chemical and a genetically susceptible individual, or intrinsic due to chemical overdose [8,17]. Differentiation of these two injury types is essential in the context of case evaluation, RUCAM, and biomarkers. 

### 3.2. Biomarkers

Biomarkers are defined by the National Institutes of Health Biomarkers Definitions Working Group as “a characteristic that is objectively measured and evaluated as an indicator of normal biological processes, pathogenic processes, or pharmacologic responses to a therapeutic or toxic intervention” [18]. In clinical medicine, biomarkers are used diagnostically to aid existing tools to establish the correct diagnosis of diseases that include liver injury caused preferentially by drugs and herbs [11,12,13,14,15,16,17]. Examples of biomarkers focus on the analysis of molecules such as RNA, DNA, and proteins in biological fluids [11,12,13,14,15,16,17,18,19]. New biomarkers to diagnose special diseases must been tested before in patients with these diseases, and their diagnoses must be verified using established diagnostic tools [14]. Most importantly, biomarkers require validation and methods that produce accurate and reproducible results [12,19]. 

## 4. Idiosyncratic Drug Induced Liver Injury 

### 4.1. Critical Issues in Clinical Settings

Idiosyncratic DILI is a rare event compared with other liver diseases [20]. This suggests that abnormal LTs in patients taking drugs are more likely due to chronic liver disease than DILI [20], which in turn explains why so many cases initially diagnosed as DILI were actually not DILI but due to causes unrelated to drugs [2,3,4,17]. This problem still haunts the LiverTox database because many assumed DILI cases, which had not been submitted to established causality assessments, are included and presented as DILI reference cases [2,3,4]. Therefore, other approaches, such as diagnostic algorithms and biomarkers, merit further attention [5,6,7].

### 4.2. Diagnostic Algorithms 

DILI’s clinical features are variable and may not contribute to a correct diagnosis at the initial presentation. Due to concerns regarding alternative causes that may confound DILI diagnoses [4,20] and to overcome diagnostic problems, sophisticated diagnostic algorithms had been implemented like the original RUCAM in 1993 [21,22]. Both, this original RUCAM [21,22] and the updated RUCAM of 2016 [8] became the most widely used algorithms for causality assessment in DILI and have been applied in more than 46,266 DILI cases published between 2014 and 2019 [17]. Currently preferred is the RUCAM version updated in 2016 [8], along with additional details [9,23]. As a liver specific, item scoring, structured and validated causality assessment method (CAM), RUCAM outperforms other CAMs on the market, which are no equally valid substitutes for RUCAM [8]. In an analysis of 46,266 RUCAM based DILI cases, requirement of additional biomarkers was not reported [17]. However, new diagnostic biomarkers could assist RUCAM in DILI assessment, but there was no diagnostic biomarker found or proposed at earlier times [14]. 

### 4.3. Diagnostic Biomarkers

Biomarkers of idiosyncratic DILI differ in their intentional use and may now best be classified as diagnostic or prognostic biomarkers [14,24]. Both biomarker types have been discussed in a variety of contexts [11,12,13,14,24,25,26,27,28,29,30,31,32,33], whereas the current analysis focuses on only diagnostic biomarkers. These must be used prior to prognostic biomarkers for DILI case evaluation since a prognosis depends on a correct diagnosis. A highly appreciated and straightforward 2014 editorial in *Liver International* on new DILI biomarkers raised the correct question as to whether they are really better, and what do they diagnose [12]. Additionally, no substantial progress has occurred despite major efforts [14,17].

#### 4.3.1. Potential Idiosyncratic DILI Biomarkers 

Scientific, regulatory, and consortia publications have focused on idiosyncratic or intrinsic DILI and diagnostic biomarkers that include microRNA-122 (microarray RNA-122), microRNA-192, CK-18 (Cytokeratin-18 full length), ccCK-18 (caspase-cleaved CytoKeratin-18), Cytokeratin-18 (fragments), GLDH (Glutamate dehydrogenase), total HMGB-1 (High Mobility Group Box), hyperacetylated HMGB-1, and ITGB3 (Integrin beta 3). Some of these are listed in Table 1 and have been discussed in the scientific literature [14,17,24,25,26,27,33]. However, on 15 April 2019, confusion emerged due to the EMA issuing a retraction note regarding various potential biomarkers listed above due to external data manipulation [17]. As early as 2016, EMA had presented online a letter of support to use several diagnostic biomarkers in clinical trials to verify or exclude liver injury cases, an approach that had created a scientific and clinical biomarker hype. The official retraction three years later was absolutely correct but caused uncertainty among DILI experts who so far used the biomarkers under consideration, not allowing final conclusions on the topic right now. 

#### 4.3.2. ITGB3

Currently, ITGB3 is the only parameter that fulfills partially the requirements of a diagnostic biomarker for idiosyncratic DILI [33], considering a variety of limitations and uncertainties (Table 1) [17]. In patients with assumed idiosyncratic DILI by diclofenac, in vitro proteomics analyses of monocyte-derived hepatocyte-like cells identified ITBG3 as a specific biomarker of DILI by diclofenac, a conclusion of the authors reached from a small study cohort of 12 cases and 10 controls [33]. Whether additional studies can confirm these primarily encouraging data remains to be established. Indeed, recommending this parameter in clinical practice at present is premature.

#### 4.3.3. Current Challenges 

Other parameters presented in the literature as biomarkers are disappointing from a clinical and regulatory view because most of them have nothing in common with potential diagnostic biomarkers (Table 1) [14,24,25]. The most relevant reasons are data presentation and interpretation incoherencies related to assumed diagnostic biomarkers and evident from rarely defined idiosyncratic DILI, missing DILI causality assessment (such as RUCAM), lack of diagnostic clinical criteria for liver injury, missing or ongoing validity testing, unknown targeted organ, liver injury and drug specificities, and unclear superiority over existing diagnostic biomarkers such as ALT and ALP. Most reports described the mechanistic background responsible for cellular or subcellular origin of the parameter under consideration and reported on conflicting data open for further discussions. Little support existed for the provided parameters listed in other publications [26,27,28,29,30,31,32,33,34]. This included a recent commentary carefully discussing biomarker options without considering new regulatory aspects of EMA (European Medicines Agency) [34], as previously discussed [17]. 

### 4.4. Proposals for Future Approaches 

Triggered by hastily presented Letters of Support from regulators and consortia, a biomarker hype emerged in 2016 following EMA’s note (EMA/423870/2016) available online until April 2019: http://www.ema.europa.eu/docs/en_GB/document_library/Other/2016/09/WC0021379.pdf. 

The biomarker hype ended with a dramatic scientific and regulatory dilemma due to EMA’s retraction of its earlier Letter of Support, as briefly outlined [17]. To overcome the present problems searching for valid diagnostic biomarkers of idiosyncratic DILI, previously published analyses and recommendations should be reconsidered [14]. 

EMA’s retraction was unexpected for DILI experts as posted on its website on 15 April 2019: https://www.europa.eu.documents/other/retraction-letter-support-drug-induced-liver-injury-dili-biomarker-ema/423870/2016_en.pdf. The website with the retraction notification remained accessible only for a short period before it was removed from the internet, concomitantly removed with the initial Letter of Support that was also not available online anymore. However, both the Letter of Support and the retraction notification have been preserved as files, kept in the private archive of the first author (RT), and ready to be provided, with details recently discussed [17]. Indeed, misconduct at one of the collaborating partner centers led to the official and correct statement of EMA no longer recommending the exploratory hyperacetylated HMGB1 isoforms in clinical studies. The EMA suggested that the overall promising nature of the other promoted biomarkers was considered highly dependent on the initially outstanding results of the later incriminated biomarker hyperacetylated HMGB1. Therefore, the EMA decided to retract its Letter of Support affecting virtually all biomarkers listed in Table 1. As a consequence, related regulatory or consortia Letters of Support previously provided by the FDA and SAFE-T (Safer and Faster Evidence-based Translation) Consortium disappeared from their websites that were previously cited in the literature [14,17,24,34]. All such conditions require new considerations. 

Among these are (1) alongside the official retractions and upcoming uncertainties, a major dilemma emerged due to overt gaps related to diagnostic biomarkers and idiosyncratic DILI, requiring new and better approaches; 

(2) regulators and consortia are well advised to now build on the knowledge of experts in the field of idiosyncratic DILI and RUCAM; 

(3) new biomarkers should be validated in patients with idiosyncratic DILI that had been assessed for causality by using the updated RUCAM as gold standard [8,35], in line with earlier recommendations [14]; 

(4) only DILI cases with a high causality grading (RUCAM score ≥6, providing a causality grading of probable or higher) should ideally be used, along with perhaps the newly developed staring system of RDCQ (RUCAM-DILI Case Quality) [17]; 

(5) high test specificities and sensitivities are required for each newly proposed diagnostic biomarker [14]; 

(6) liver injury must be major and evaluations must follow certain current criteria: serum activities of LTs, namely ALT of at least 5 x ULN (upper limit of normal) and/or ALP of hepatic origin and of at least 2× ULN. It is important to recognize this ALT threshold early at time of initial presentation and to remove all cases with minor and or usually reversible liver injury [8,17]; 

(7) organ specificity (liver), usually provided by ALT but can be improved by GLDH; 

(8) drug specificity; 

(9) large size of cohort; 

(10) to be evaluated in humans and not using experimental tissues or animal models; 

(11) search for urinary biomarkers is encouraged; and 

(12) overall, a prospective study protocol is required [8], excluding search for potential diagnostic biomarkers of intrinsic DILI and liver specificity as illustrated above (Table 1). 

## 5. Intrinsic Drug Induced Liver Injury

### 5.1. Critical Issues in Clinical Settings 

Intrinsic DILI is caused by drug overdose, with paracetamol (acetaminophen, *N*-acetyl-p-aminophenol, APAP, Tylenol^®^) as the most common drug responsible [5,6,7,36,37,38,39,40,41,42,43]. Occasionally, other drugs like amiodarone, anabolic steroids, cyclosporine, valproic acid, heparins, nicotinic acid, methotrexate, and tacrine have also been assumed to cause intrinsic DILI, with rare acute intoxications. Liver injury in these cases appears due to prolonged use or unrecognized, drug unrelated chronic liver diseases, and would falsely be labelled as DILI if RUCAM was not applied for causality. Human intrinsic DILI due to paracetamol is reproducible in animal models with the advantage of providing insights into mechanistic pathways of the liver injury that researchers can translate to humans [38,39,40]. Not unexpectedly, clinical treatment using paracetamol is under major discussion following a Cochrane statement that it is largely ineffective [44,45]. Other interesting clinical aspects related to paracetamol are discussed below in terms of liver injury.

### 5.2. Diagnostic Algorithms

To establish the diagnosis of intrinsic DILI, identical thresholds for ALT and ALP are applied as described above for idiosyncratic DILI [8]. Again, the diagnostic algorithm includes clinical evaluation, followed by the use of the updated RUCAM to verify or disregard causality for paracetamol [8,40,42]. RUCAM could help decrease the high case numbers of undetermined causes in cohorts of ALF (acute liver failure) [43]. In addition, RUCAM is a valuable tool for deciphering concomitantly used drugs as a potential cause of liver injury [8] as well as in cases of acute on chronic liver failure due to paracetamol [42]. 

RUCAM based analyses classified three different liver modifications by paracetamol use: (1) it confirmed the usual paracetamol-overdose induced intrinsic DILI, (2) established following normal dosed paracetamol not only rare idiosyncratic DILI in analogy to many commonly used drugs but (3) also rare liver adaptation or tolerance, a mild form of hepatic involvement due to metabolic modification or tolerance [40]. Therefore, the spectrum of liver involvement by paracetamol is much broader than previously assumed.

### 5.3. Diagnostic Biomarkers

Chances to uncover diagnostic biomarkers are better if high amounts of a drug cause intrinsic DILI as compared with idiosyncratic DILI, for which a drug is prescribed in normal dosages [14]. Numerous mechanistic pathways have been explored to determine potential diagnostic biomarkers [5,6,7,28,46,47,48,49,50]. However, questions have been raised as to whether specific diagnostic biomarkers in patients with suspected DILI by paracetamol are really needed clinically if the case evaluation included the common medical practice of using RUCAM [40]. In particular, microRNAs including microRNA-122 have been evaluated in experimental liver injury and in human intrinsic DILI caused by paracetamol, but their diagnostic value as diagnostic biomarker remains unclear due to lack of information on the method that was used to assess causality [40]. Some groups encouraged their use at least partially [5,13,28,47,48,49], while others were more cautious and refrained from mentioning these tools [50]. It is also clear that microRNA-122 is only a simple marker of liver injury (Table 1), assessable in the capillary blood of patients with intrinsic paracetamol induced DILI, whereby causality for paracetamol was ascertained using the updated RUCAM [51]. 

Paracetamol protein adducts in the serum and urinary biomarkers have also been considered of some clinical value in suspected DILI cases possibly initiated by paracetamol overdose [5,6,7,52,53,54,55,56,57]. It seems, however, that urinary mercapturic or GSH-NAPQI (*N*-acetyl-p-benzoquinonimine) adducts are the preferred biomarkers as they appear before serum protein adducts are detected. 

Diagnostic biomarkers in paracetamol induced DILI remain an issue [5,6,7,40,52,53,54,55,56,57] due to their inability to solve the problems of alternative diagnoses commonly confounding the DILI [20]. Cases of DILI by paracetamol used for testing new biomarkers are likely not correctly assessed by a validated CAM such as RUCAM, an omission that substantially decreases the power of the tested biomarker [14,40]. 

#### 5.3.1. Current Challenges

Current conditions of undetermined causes in cohorts of ALF are unsatisfactory due to limited use of RUCAM and lack of appropriate diagnostic biomarkers. Paracetamol is a frequent cause of ALF, but this likely remains unrecognized in a number of suspected cases. Correct diagnoses of DILI by paracetamol are essential for early treatment options, such as the use of NAC to replenished exhausted hepatic glutathione levels. Early and correct diagnosis will help reduce the need of liver transplantation. 

#### 5.3.2. Proposals for Future Approaches 

Future patients with incipient liver injury should be evaluated for current paracetamol use. This can best be achieved by improved clinical screening protocols and the use of RUCAM.

## 6. Herb Induced Liver Injury 

In most published cases of HILI, an idiosyncratic mechanism can be assumed following herbal use in recommended daily doses. A few herbs cause dose dependent liver injury considered as intrinsic HILI, based on the criterion of acute intoxicating ingestion. 

### 6.1. Critical Issues in Clinical Settings 

Worldwide occurrence of HILI is likely lower as compared with DILI [58,59,60,61] although in some Asian regions a reversed constellation is evident with up to 55% ascertained as HILI versus 45% as DILI [59,60]. For HILI by traditional Chinese Medicines (TCM), the most commonly reported regions were in Hong Kong, Korea, Japan, and the United States, whereas reports originating from European countries were unexpectedly scarce [16]. However, problematic are reports that include both DILI and HILI cases without a clear separation of the two groups essential for individual case cohort characterization [17]. 

There is also the concern of using the fragile term “dietary supplements” (DS) [61,62,63]. For instance, herbal products containing green “tea”, are they really supplementing the diet or serving as a beverage? A large online table presented multiple case reports of HILI and provided evidence that scientists from virtually all countries contributed case details, whereby HILI was described following use of regulatory approved herbal drugs, non-approved herbal medicines, herbal products in general, and herbs contained in so called DS [62]. Perhaps some DS are better replaced by the term herbal products because supplementation in humans is not the primary aim of all DS products although this term is liked by potential or current DS producers to increase profit, but some practitioners and researchers feel that physicians should not promote the DS industry [63]. 

### 6.2. Diagnostic Algorithms

Suspected HILI cases are best assessed as described above for idiosyncratic DILI: clinical evaluation first and followed by causality assessment using the updated RUCAM [64]. This approach is also recommended in pharmacovigilance settings mentioned in the guidelines for the diagnosis and management of HILI, published by the China Association of Chinese Medicine [65], all in line with corresponding recommendations proposed by the Chinese Society of Hepatology (CSH) for DILI [66]. The broad use of RUCAM for HILI cases is also evidenced by a variety of international registries and regulatory agencies in the Asian region and countries elsewhere [16]. Otherwise, the list of RUCAM based HILI cases is long [16], in addition to reports published until 2015 [8]. To further underscore this point, a few selected examples of other RUCAM based HILI cases are mentioned [67,68,69,70,71,72,73,74], referring to green tea extracts [67], *Polygonum multiflorum* [68,69], TCM in Germany [70], and cohort studies or case series [71,72,73,74].

### 6.3. Diagnostic Biomarkers

Since most HILI cases have an idiosyncratic background, valid biomarkers in large numbers cannot be expected [16]. Liver injury caused by green tea extracts shows dose dependent features but diagnostic biomarkers were not described [67]. Dose dependency can be assumed for HILI caused by phytochemicals derived from germander (*Teucrium chamaedris*) [17,75,76,77,78,79], *Polygonum multiflorum* [68,69,80,81], and plants containing unsaturated pyrrolizidine alkaloids (PAs) [15,82,83,84,85,86,87,88,89,90]. For these herbs, diagnostic biomarkers are known and in clinical use although test validation was rarely provided [17]. 

Liver injury by germander is dose-dependent and can easily be recognized using a specific diagnostic biomarker [17,75,76,77,78,79]. Germander components undergo microsomal oxidation via CYP isoform 3A [75,76,77,78,79]. Indeed, anti-microsomal epoxide hydrolase autoantibodies have been found in the sera of patients who drank germander tea for a long period of time. 

In experimental liver injury by *Polygonum multiflorum*, nine biomarkers were identified using a pathway analysis, which indicated some metabolic involvement of pathways related to lipids, amino acids, and bile acids, but this was not evaluated in patients with liver injury [81]. In clinical studies using prospectively RUCAM for causality assessment in patients with HILI caused by *Polygonum multiflorum*, a sophisticated but complex evidence chain-based causality identification algorithm was used that also included metabolomics analyses [68]. 

Liver injury by plants containing PAs is commonly classified as hepatic sinusoidal obstruction syndrome (HSOS) [15,16,82,83,92,93]. A new diagnostic biomarker of HSOS caused by PAs is now available and was applied in patients who also were assessed for causality using RUCAM [82,83,92,93]. The new diagnostic biomarker is a sensitive and specific assay enabling to detect reactive pyrrole-protein adducts in the serum of patients with PA related HSOS; it is most commonly applied in China due to high frequency of appropriate cases [82,83,92,93]. In Germany, regulators assumed liver injury under a prophylactic therapy of *Petasites hybridus* for migraine because previous herbal drugs contained small amounts of PAs that have now been removed due to a refined production technique; in more detail, RUCAM based HILI case assessment showed no causality for this herb, associated with lacking HSOS by liver histology evaluation and missing use of PA specific biomarkers [84]. 

#### 6.3.1. Current Challenges 

Clinical and regulatory approaches currently focus on diagnostic algorithms to improve the diagnosis of HILI. For causality assessment of HILI, RUCAM has a good historical background and is used worldwide, whereas the number of diagnostic biomarkers is limited, not allowing substantial support of RUCAM. Although most HILI cases have correctly been evaluated for causality, listings contained in databases or scientific publications would be greatly improved if all HILI cases, as well as all DILI cases, received prior evaluation using RUCAM or biomarkers. Some examples are referenced including earlier reports from our group [16,35,62,80,81,82,83,84,85,86,87,88,89,90,91,92,93,94].

Alternative diagnoses can be found during assessment of HILI cases and are crucial [95], because initially assumed HILI cases were not HILI as reported by some researchers providing a few examples [96,97,98,99,100,101,102]. Available diagnostic biomarkers are of little help as opposed to RUCAM with its algorithms specifically prepared to search for alternative causes.

#### 6.3.2. Proposals for Future Approaches 

For assessing HILI cases, new approaches are needed to search for diagnostic biomarkers that are validated using an established gold standard such as the updated RUCAM. For this purpose, ideally only HILI cases should be used with at least a probable causality grading, meaning a RUCAM score ≥6. Rather than dealing with causality problems, a better approach is reducing assumed HILI cases by strictly following the benefit–risk balance because many herbal products lack efficacy while adverse reactions including liver injury are well known. More specifically, randomized clinical trials are urgently needed [103].

## 7. Alcoholic Liver Injury 

### 7.1. Critical Issues in Clinical Settings

Alcoholic liver disease (ALD) is well characterized and represents a worldwide health problem [104,105,106,107,108,109,110,111,112,113,114,115,116,117,118,119,120,121]. Main issues relate to early recognition of the disease and clear differentiation from other frequent acute and chronic liver diseases. 

### 7.2. Diagnostic Algorithms

ALD is basically a preventable disorder requiring early diagnosis to achieve alcohol abstinence and impede development of serious stages of ALD like alcoholic hepatitis, cirrhosis, and hepatocellular carcinoma [107,108]. The respective diagnostic algorithms include most importantly intuition of physicians to recognize a patient with a risk of alcohol abuse. Questionnaires may be helpful [116,117], along with ultrasound examination of the liver to search for fatty liver, and laboratory tests [107,108,116,117,118]. 

### 7.3. Diagnostic Biomarkers 

Various diagnostic biomarkers are under discussion that can help search for individuals with severe alcohol abuse [117,118]. Respective laboratory data show variable percentages of sensitivity: carbohydrate-deficient transferrin (CDT; 63%), gamma-glutamyltransferase (GGT; 58%), mean corpuscular volume of erythrocytes (MCV; 45%), aspartate aminotransferase (AST) (47%); ALT; 50%), and GGT + CDT (90%) [117]. However, and as expected, the specificity of these parameters is low. Known for a long time in clinical practice, high serum GGT activities in alcoholic patients with falling tendency due to abstinence from alcohol use are of clinical interest [117]. In addition, a serum ratio of AST/ALT >2.0 provided diagnostic clues in ALD and can be considered as a valuable diagnostic biomarker [117,120]. Similarly, serum GLDH activity may be a good marker for alcoholism, with a 5-fold increase in patients with alcoholic fatty liver AFL, which was significantly different from controls and showed little variability [122,123]. Further studies are required to assess the validity of microRNA-182 as biomarker in ALD [124]. 

Other potential biomarkers are closely related to the metabolism of ethanol via the microsomal ethanol-oxidizing system (MEOS) with CYP 2E1 as its major component and the associated production of acetaldehyde and ROS [117,118,119,125,126,127,128,129,130]. This explains why antibodies against acetaldehyde proteins are found in the serum of individuals with an alcohol abuse. Also exciting are studies on circulating blood exosomes [117,131]. In the blood of patients with a history of alcohol use and animals exposed to binge alcohol or repeated doses, extracellular vesicles containing CYP isoforms were detected, namely CYP 2E1, 2A6, 1A/2, and 4B in patients, and CYP 2E1, 2A3, 1A/2, and 4B in animals [131]. 

### 7.4. Current Challenges 

Early diagnosis of ALD is essential requiring various strategies including the use of a sensitive and specific biomarker. Present available tests are acceptable but need further refinement in terms of evaluating validity. 

### 7.5. Proposals for Future Approaches 

Given the large group of patients with an alcohol related liver problem, compared with other liver diseases, it should be easy to search for a few additional reliable diagnostic biomarkers. 

## 8. Conclusions 

Patients with suspected liver injury caused by exogenous substrates such as drugs, phytochemicals, or alcohol require a thorough clinical evaluation that includes strict diagnostic algorithms. Results of additional biomarkers in support of the diagnosis are variable and occasionally debated. Robust results of biomarkers are available for intrinsic liver injury due to intoxicating acetaminophen and phytochemical PAs. For the rare idiosyncratic DILI caused by most drugs in susceptible individuals, a variety of biomarkers is under critical discussion due to EMA’s retraction of official support initially provided by EMA itself. Promotion of ITGB3 as potential diagnostic biomarker of idiosyncratic DILI seems currently premature due to some evident uncertainties. New biomarkers are desirable for idiosyncratic DILI, but they must be validated using RUCAM based DILI cases with a high causality grading as gold standard. Whether promotion of diagnostic biomarkers in liver injury cases is feasible by upcoming artificial intelligence approaches remains to be seen.

## Figures and Tables

**Table 1 ijms-21-00212-t001:** Selected potential diagnostic biomarkers of idiosyncratic liver injury that have now mostly been retracted by EMA.

Parameter	Comments	Reference (First Author)
MicroRNA-122	● Released from damaged hepatocytes; validation ongoing.	Fontana [25]
	● Previously proposed by EMA for early recognition of liver injury in trials, reflecting liver cell necrosis.	Teschke [14]
	● Functions in the context of hepatocyte differentiation, hepatitis C virus infection, and lipid metabolism, and but is not DILI specific, lacking data of specificity and sensitivity for idiosyncratic DILI.	Church [24]
	● Recommended by regulatory Letters of Support.	Church [24]
	● Considered as liver specific but this biomarker is also produced by non-hepatic cancerous cells, thereby questioning liver specificity characteristics.	Church [24]
	● Previously proposed by SAFE-T consortium: Liver specific, early marker possibly preceding ALT on a temporal scale. Reported as sensitive DILI marker in multiple clinical studies, but robustness of specificity and sensitivity data for idiosyncratic DILI not provided.	Teschke [14]
	● Considered as promising biomarker of DILI with increased hepatic specificity.	Antoine [26]
	● Becomes a promising candidate for responding to the need for more specific and sensitive biomarkers for DILI.	Krauskopf [27]
	● Preliminary data for idiosyncratic DILI, also low case number.	Liu [13]
MicroRNA-192	● Liver specific release from damaged hepatocytes; validation reported as ongoing.	Fontana [25]
GLDH	● Revival of an old diagnostic biomarker, considered liver specific without providing validation, suggested potential utility in DILI as described by regulatory Letters of Support but lack of DILI specificity.	Church [24]
	● Previously proposed in EMA Letter of Support: Parameter of liver cell necrosis.	Teschke [14]
Cytokeratin-18 (full length)	● Sensitive biomarker for necrotic cell death, but not liver disease specific.	Fontana [25]
	● Classified as prognostic biomarker, not as diagnostic biomarker.	Church [24]
Cytokeratin-18 (fragments)	● Marker of caspase cleaved proteins in apoptotic cell death, noted in individuals with ongoing apoptosis, not liver specific.	Fontana [25]
Total HMGB-1	● Sensitive biomarker for necrotic cell death, but not liver disease specific.	Fontana [25]
	● Previously proposed by SAFE-T: Regarding origin of biomarker, detectable in almost all tissues, not specified for idiosyncratic DILI.	Teschke [14]
	● Previously proposed by EMA for early detection of liver injury in trials as a biomarker of liver cell necrosis.	Teschke [14]
Acetylated HMGB-1	● Innate immune activation marker, acetylation requires mass spectroscopy, nevertheless, parameter is listed under the segment of liver injury markers.	Fontana [25]
Integrin beta 3 (ITGB3)	● Lacking a prospective study protocol, ITGB3 was retrospectively tested in 16 patients with undefined liver injury criteria under treatment, a not further described clinical causality assessment including RUCAM (updated version?). Common case narratives and individual core elements for each patient were not provided. Data on specificity and sensitivity were not presented although the test was described as drug specific biomarker.	Dragoi [33]
	● Reported causality assessment likely included the use of the updated RUCAM, however, all cases were wrongly classified by the authors as having for DILI by diclofenac a likelihood of at least “highly likely” (not a causality grading of RUCAM!) but RUCAM scores were published only as median of 8 and a range of 7–9, while a RUCAM score ≥9 qualifies for a highly probable causality grading. Uncertainty remains also regarding the expression of “at least” that would imply tentative higher causality gradings that in fact do not exist and were not published. At another place, causality likelihood was described as definite and highly likely, both terms are not those of RUCAM and must have been derived from somewhere else. Applied reexposure criteria remained unreported. Certainly, it is a preliminary though encouraging approach with a small number of patients, using an in house developed test and now published by in house investigators.	Teschke [17], Dragoi [33]
	● Recommendation to use would be premature as outlined above.	Teschke [17]

Note: With the exemption of ITGB3 but for virtually all parameters above, EMA retracted its earlier Letter of Support to be used in trials due to a fraudulent external study, leaving the primarily assumed diagnostic biomarkers in uncertainly as briefly discussed and referenced elsewhere [17]. Abbreviations: ALT, Alanine aminotransferase; ccCK, caspase-cleaved CytoKeratin; CK, CytoKeratin; DILI, Drug induced liver injury; EMA, European Medicines Agency; GLDH, Glutamate dehydrogenase; HMGB, High Mobility Group Box (protein); microRNA, microarray RNA; RUCAM, Roussel Uclaf Causality Assessment Method.

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
