# Peer review of "Diagnostic Biomarkers in Liver Injury by Drugs, Herbs, and Alcohol: Tricky Dilemma after EMA Correctly and Officially Retracted Letter of Support"

_ijms, 2019, doi:10.3390/ijms21010212_

Round 1

Reviewer 1 Report

My remark regarding the correction of paracetamol-protein adducts sentences of the authors should be seen as a correction regarding a misleading information of the authors on paracetamol pharmacokinetics. Still, NADPQI-adducts cannot be used at this stage as biomarkers of liver injury in the clinical use as they are not ready to determine as to corroborate a rapid clinical decision (HPLC or LC_MS is required) or no clear correlation on its levels and hepatic injury occurs.

Please read and comment on Biomarkers for risk assessment in paracetamol hepatotoxicity

William Bernal Lancet Gastroenterology 2017. It gives straightforward info regarding new markers of paracetamol toxicity in clinical practice.

For this reviewer the introduction of herbal, CCL4 and so on is confusing, when this should be a review regarding the authors opinion on the EMA decision.

The authors section named 6 is repeated. The numbers are wrong.

Author Response

Dear Ms. Xiang and Reviewers,

I hope the word count is now ok, or how many words have to be deleted now in the revised version? Reviewers asked to add several sentences (not important in my view) which of course increases the word count but this is not my fault. If further reduction is needed, please let me know. Actually, the invitation letter that I received did not contain a word number limitation. I just checked the word count of the originally submitted version without abstract, references and table. Word count of the text was only 5721. So the word count of the revised version must be much lower as I deleted many sentences. Being not a PC expert, I have no software on my PC to assess the word number after deletion on the many words and sentences. If this is still needed, please have the crucial counts be done by one of the PC experts of the editorial board.

Thank you for your valuable comments to improve the quality of our invited paper. According to the Editor’s comment, we condensed the paper substantially.

Please find our answers for the reviewers in color below: 

Comments and Suggestions for Authors

The work of Teschke et al seemed of great importance in the field and both the title and abstract (although this last sometimes confusing) were appealing to an interesting discussion on hepatic toxicity diagnosis or at least an open discussion on the pros and cons of EMA decision. Still, the manuscript per see was disappointing. No clear structure is given and the authors should revise the aims of the work.

Aims of the work were expanded: 74-76.

First, the authors should clearly state the definition of biomarkers and possibly select only biomarkers of effect and exposure to the paper. The specificity of those can be discussed case by case, but not in a random form. Moreover, it is not clear to me if the authors aim to discuss idiosyncratic DILI (as it seems in the abstract) or intrinsic and mostly dose dependent DILI, as it often happens in the text. In fact, the xenobiotics used are mainly responsible for intrinsic DILI and exposure biomarkers exist although they are not fast and easily available to the clinicians (When the authors mention paracetamol and protein adducts, the authors engage in a serious error; before protein adducts appear, mercapturic or GSH – NAPQI adducts appear, even in urine; the detection of protein adducts may not bring anything more substantial to the table that classic ALT with easy paracetamol detection does not give already).

Definition of biomarkers was expanded: 103-105. If available, the specificity of the biomarkers was mentioned in sections related to drugs, herbs, alcohol, and carbon tetrachloride. Thank you for suggesting clarification. Accordingly, section of paracetamol biomarkers was rewritten more critically: 267-291.

Table 1 is just a simple and not discussed form of setting markers already nor supported by EMA. In fact, the main author already published in 2016 defending the RUCAM system and I am sure can give a more critical analysis of the EMA retraction. EMA’s decision are not without controversy and not always agree with other respectable agencies. In other situations, EMA has removed support to some drugs based on biased assays, whereas FDA maintained the support, since authors of those studies retracted the statements or were retracted (namely Dexarazone). Mostly, most of FDA black boxes on drugs are due to hepatotoxic drugs, thus valid biomarkers (either of exposure, but mostly of effect) most be endorsed in large clinical studies).

Regarding Table 1 and EMA’s retraction, expanded comments are given: 151-157, 188-211. Table 1 should be seen as a short information; due to EMA’s retraction that invalidates biomarkers initially suggested by EMA, only ITGB3 merits detailed discussion. Further details seem currently not justified and speculative.

Minor issues:

When the authors write statements like this: ‘There is not much to add to this Editorial, except perhaps the fact that no substantial 140 progress has occurred despite major efforts [14,17].’, we also doubt the novelty of the work.

Sentences like this are rhetorical: Given this 237 published opinion on acetaminophen (APAP) hepatotoxicity - isn't it time for APAP to go away? [45], 238 other interesting clinical aspects related to paracetamol are discussed below in terms of liver injury.

In fact, several of the drugs used today would not survive clinical or even pre-clinical assessment, but they are in the market

The statements on the Editorial and on APAP were now deleted or modified: 143-144; 247-248.

Why is not non-alcoholic liver disease not mentioned?

NALD was not included because it is not caused by an exogenous chemical.

Comments and Suggestions for Authors

The present manuscript by Rolf Teschke et al. is a comprehensive review discussing the role of emerging biomarkers to diagnose drug-induced and herb-induced liver injury with an especial focus on a recent position of the EMA questioning their supporting evidence. The manuscript is informative. The main limitation is the excessive length which makes it hard to read.

Thank you for encouragement and proposals to improve the paper.

Length was heavily reduced as suggested.

The authors are invited to consider the following comments:

- The manuscript is too descriptive and sometimes redundant with some ideas such as the caveats to diagnose DILI, the relevance of differential diagnosis, the RUCAM approach… The authors should consider shortening significantly some sections to make it more straightforward.

- Table 1 is mainly composed by unstructured text. For instance, it can be read as follows: “Reported causality assessment likely included the use of the updated RUCAM, however, all cases were wrongly classified by the authors as having for DILI by diclofenac a likelihood of at least “highly likely” (not a causality grading of RUCAM!) but RUCAM scores were published only as median of 8 and a range of 7-9, while a RUCAM score ≥9  qualifies for a highly probable causality grading”. And this paragraph continues for at least 10 more lines. The structure of table 1 could be improved by limiting the information of each biomarker to practical clinical aspects: initial description, external validation, consistent thresholds, accuracy, availability…

Table 1 provides a few details only, expansion seems currently not warranted due to EMA’s retraction, invalidating biomarkers initially suggested by EMA. Consequently, only ITGB3 merits detailed discussion.

- An alternative structure for the manuscript could be considered as follows: 1) Caveats to diagnose DILI/HILI; 2) The RUCAM approach; 3) Biomarkers of DILI/HILI (including table 1); 4) Recent recommendations by EMA; 5) Future directions; 6) Conclusion.

- Alcohol derived liver toxicity seems unrelated to the topic (DILI/HILI). I would recommend removing this section.

Thank you for new structural considerations. We prefer keeping the current structure as not only DILI and HILI have to be considered. As an exogenous compound, alcohol fits to drugs and herbs, but section was condensed.

Kind regards,

Rolf Teschke

Reviewer 2 Report

The authors have made significant efforts to shorten the manuscript and to improve structure as suggested. Most of my initial concerns have been adequately addressed. 

Author Response

(The authors gave the same response as above.)

Round 2

Reviewer 1 Report

The authors are struggling to understand the reviewers´ quest for clarity and objectivity. Please take note for some suggestions that I believe will increase the impact of the manuscript.

Please correct Typo: Germany, Line 9

Line 73-77 are more clear in the objective of the study than the abstract

Since all biomarkers in table were removed recommendation, why that information is not placed on the title or footnote of the table instead of the repeating ‘Recommendation retracted by EMA as referenced elsewhere. ’

The hope of the authors that ITGB3 is to be considered a valid biomarker is very poorly corroborated in the text.

I insist on removing at least CCl4. It is out of focus and if authors complain that the reviewers ask to add sentences and they exceed length, this is a good way to shorten the paper. It is completely off the objective.

In fact, CCL4, ethanol or herb toxicity is not mentioned in the abstract. Moreover, I do not believe that the use of CCL4 as a model to produce liver injury in selected animal models is relevant for biomarkers of human liver disease. The diagnosis algorithms or biomarkers placed by the authors represent no novelty when compared to the other data discussed.

Author Response

Dear Ms. Xiang and Reviewer,

Thank you for your valuable comments to improve the quality of our invited paper. Please find our answers for the reviewer in color below: 

The authors are struggling to understand the reviewers´ quest for clarity and objectivity. Please take note for some suggestions that I believe will increase the impact of the manuscript. Thank you, we understand.

Please correct Typo: Germany, Line 9. Done.

Line 73-77 are more clear in the objective of the study than the abstract. Your comment is appreciated.

Since all biomarkers in table were removed recommendation, why that information is not placed on the title or footnote of the table instead of the repeating ‘Recommendation retracted by EMA as referenced elsewhere. ’ Was now condensed as requested: Table 1, L159,160,163.

The hope of the authors that ITGB3 is to be considered a valid biomarker is very poorly corroborated in the text. Was now toned down on L 174-175.

I insist on removing at least CCl4. It is out of focus and if authors complain that the reviewers ask to add sentences and they exceed length, this is a good way to shorten the paper. It is completely off the objective. Done.

In fact, CCL4, ethanol or herb toxicity is not mentioned in the abstract. Alcohol and herb toxicity was now included in the abstract: L18. Moreover, I do not believe that the use of CCL4 as a model to produce liver injury in selected animal models is relevant for biomarkers of human liver disease. The diagnosis algorithms or biomarkers placed by the authors represent no novelty when compared to the other data discussed. Section of CCl4 and related text were now completely removed: L 483-514, in the title, abstract, and elsewhere in the text. Ref 138-140 were removed.

Thanks again for your patience and cooperation.

Kind regards,

Rolf Teschke